# Frontier Review of the Molecular Mechanisms and Current Approaches of Stem Cell-Derived Exosomes

**DOI:** 10.3390/cells12071018

**Published:** 2023-03-26

**Authors:** Liang-Yun Chen, Ting-Wan Kao, Chang-Cyuan Chen, Noreen Niaz, Hsin-Lun Lee, Yu-Hsin Chen, Chia-Chun Kuo, Yao-An Shen

**Affiliations:** 1Department of Pathology, School of Medicine, College of Medicine, Taipei Medical University, Taipei 110301, Taiwan; 2Graduate Institute of Clinical Medicine, College of Medicine, Taipei Medical University, Taipei 110301, Taiwan; 3International Master/Ph.D. Program in Medicine, College of Medicine, Taipei Medical University, Taipei 110301, Taiwan; 4Department of Radiation Oncology, Taipei Medical University Hospital, Taipei 110301, Taiwan; 5Department of Radiology, School of Medicine, College of Medicine, Taipei Medical University, Taipei 110301, Taiwan; 6Taipei Cancer Center, Taipei Medical University, Taipei 110301, Taiwan; 7School of Health Care Administration, College of Management, Taipei Medical University, Taipei 110301, Taiwan; 8Ph.D. Program for Cancer Molecular Biology and Drug Discovery, College of Medical Science and Technology, Taipei Medical University and Academia Sinica, Taipei 110301, Taiwan

**Keywords:** stem cell, exosome, stem cell-derived exosome, regenerative medicine, cell-free therapy

## Abstract

Exosomes are effective therapeutic vehicles that may transport their substances across cells. They are shown to possess the capacity to affect cell proliferation, migration, anti-apoptosis, anti-scarring, and angiogenesis, via the action of transporting molecular components. Possessing immense potential in regenerative medicine, exosomes, especially stem cell-derived exosomes, have the advantages of low immunogenicity, minimal invasiveness, and broad clinical applicability. Exosome biodistribution and pharmacokinetics may be altered, in response to recent advancements in technology, for the purpose of treating particular illnesses. Yet, prior to clinical application, it is crucial to ascertain the ideal dose and any potential negative consequences of an exosome. This review focuses on the therapeutic potential of stem cell-derived exosomes and further illustrates the molecular mechanisms that underpin their potential in musculoskeletal regeneration, wound healing, female infertility, cardiac recovery, immunomodulation, neurological disease, and metabolic regulation. In addition, we provide a summary of the currently effective techniques for isolating exosomes, and describe the innovations in biomaterials that improve the efficacy of exosome-based treatments. Overall, this paper provides an updated overview of the biological factors found in stem cell-derived exosomes, as well as potential targets for future cell-free therapeutic applications.

## 1. Introduction

Cell-based therapies, such as stem cell transplantation, have long been regarded as key players in the field of regenerative medicine. In the last decade, research into the mechanism of action of cell-based treatments has revealed that intercellular signaling mediates their therapeutic effects. In other words, the biologically active molecules generated by therapeutic cells, either stem cells or non-stem cells, are the actual driving force behind their effect. Because of this, a fresh page was turned in the history of cell-free therapy.

Cell-free therapy is a novel way to explore the therapeutic effects of cell-based therapy without the need for cellular transplantation. In cell-free therapy, cells are utilized as a source of biologically effective molecules, rather than as a direct therapeutic agent. Research and clinical studies on cell-free therapy have demonstrated better tolerance and comparable, or even superior, effects to cell-based therapies in several fields of applications, [1]. Cell-free therapies utilize products secreted by cells. Materials for cell-free therapy can encompass secretomes of cells and extracellular vesicles (EVs). It is clear that, biologically, there are three types of EVs, based on their distinct release mechanism, but there is no consensus on the criteria for categorizing these vesicles experimentally. It is known that microvesicles (MVs), or ectosomes, are formed by directly budding through the cell membrane, while exosomes, or multivesicular bodies emerge by directly budding into the endosomes. Depending on the multiple vesicle body type, content is either digested by lysosomes, or released as exosomes via cell membrane fusion. Thirdly, apoptotic bodies are formed from apoptosis. The formation of apoptotic bodies is thought to occur either through the separation of membrane blebs or through the formation of apoptopodia during apoptosis. It remains to be discovered how these different routes differ in terms of their courses and roles [2].

This review focuses on the novel applications of stem cell-derived exosomes in regenerative medicine, and elaborates on the diverse mechanisms underlying their activities.

### 1.1. Characteristics of Exosomes

Exosomes are EVs between 30 and 160 nm in size that originate from the endosomal network of a cell. Exosomes are able to transport substances such as proteins, lipids, DNA, mRNA, miRNA, and long non-coding chain RNAs (lncRNA) to recipient cells. Depending on the cells of origin and environmental conditions, exosomes serve distinct functions, and are currently recognized as essential for cell–cell communication, regulation of cellular physiology, and reprogramming of cell behavior. In humans, exosomes have been identified in most tissues and body fluids, indicating their biological significance. Extensive research has been focused on the prospective use of exosomes in the treatment of numerous diseases throughout the past decade [3].

### 1.2. Current Application of Mesenchymal Stem Cell-Derived Exosomes

Among the various exosome-based therapeutics, mesenchymal stem cell-derived exosomes (MSC-Exos) have gained great attention due to their diverse applicability in regenerative medicine. As one of the multipotent progenitor cells, mesenchymal stem cells (MSCs) are capable of self-renewal and differentiating into numerous cell lineages, including adipocytes, chondrocytes, and osteoblasts. In addition, MSCs can be readily isolated, and exhibit relatively limited immunogenicity. These advantages have sparked the development of MSC-based therapeutics, which have been widely adopted in animal disease models and clinical trials in humans. Nonetheless, concerns about carcinogenesis and immunostimulation have impeded the practical deployment of MSC-based therapy, notwithstanding the fact that they are uncommon [4,5].

From this background, MSC-Exos emerged as a preferred therapeutic option, since they are free from the safety concerns ensuing from cell administration. Moreover, they can be readily prepared into off-the-shelf products after sterilization, which is unachievable with MSCs [6,7]. Aside from their favorable clinical applicability, the capacity of MSC-Exos to affect cell proliferation, migration, anti-apoptosis, anti-scarring, angiogenesis, chondrogenesis, neurogenesis, and immunomodulation has been evidenced by an accumulating amount of research [8,9,10]. In addition, the creation of an anti-inflammatory microenvironment by MSC-Exos enables accelerated tissue repair and regeneration [11]. Collectively, these biological capabilities empower MSC-Exos to become a promising therapeutic tool in regenerative medicine for a wide range of diseases.

However, although a substantial number of preclinical experiments have demonstrated the therapeutic effect of MSC-Exos in vitro and in animal models [12,13,14], ongoing clinical studies are few. In other words, exosomes are promising, but do not yet have a practical medicinal application. This is most likely due to their high cost, as a result of their disease-and/or patient-specific nature [15] and the fact that certain mechanisms behind their action remain hidden.

The purpose of this review is to describe the molecular mechanisms underlying the regenerative potential of cell-free therapies, with particular emphasis on MSC-Exos. The modes of action are addressed based on different fields of clinical application. Moreover, innovative biomaterials and preparation techniques that improve the effectiveness of exosome-based treatments are highlighted.

## 2. Molecular Mechanisms of Exosomes

### 2.1. Musculoskeletal Regeneration

With the aging of the global population, the incidence of musculoskeletal diseases is steadily increasing [16]. Exosomes have gained great attention in musculoskeletal research due to their regenerative potential and minimally invasive nature (Figure 1).

#### 2.1.1. Sports Medicine and Musculoskeletal Rehabilitation

In contrast to the prevalent application of platelet-rich plasma (PRP) in clinical settings, the development of exosome-based therapies is still preliminary, and studies are largely preclinical. Recent research has gradually demonstrated the therapeutic efficacy of exosomes in musculoskeletal healing [17,18]. Exosomes derived from adipose-derived stem cells (ADSC-Exos), for example, have been shown to promote human rotator cuff healing by inhibiting muscle atrophy and degeneration, while also improving the histological properties of the torn tendon [19]. This could be accomplished by increasing AMP-activated protein kinase (AMPK) signaling, and thus inhibiting Wnt/β-catenin activity [20]. In addition, bone mesenchymal stem cells (BMMSCs)-derived exosomes (BMMSC-Exos) have been shown to stimulate the proliferation and differentiation of tendon stem/progenitor cells in vitro, increase the expression of mohawk, tenomodulin, and type I collagen, enhance the proliferation of local tendon stem/progenitor cells in vivo, and promote the mechanical properties of neotendons in the defect area of rat patellar tendons [21].

Furthermore, a potential treatment for tendon–bone regeneration was proposed, utilizing exosomes derived from genetically modified BMMSCs. Scleraxis is a transcription factor that is believed to play an active role in tendon–bone repair. A study by Feng et al. demonstrated that local injection of miR-6924-5p-rich exosomes, derived from scleraxis-overexpressing PDGFRα (+) BMMSCs, leads to reduced osteolysis and improved healing strength [22] (Figure 1).

#### 2.1.2. Osteoarthritis

The treatment of osteoarthritis (OA) has been challenging due to the low potential for spontaneous healing of cartilage tissue [23].

In the application of exosomes in OA, several studies have proposed possible mechanisms behind its effect. It is well recognized that the proinflammatory cytokine IL-1β is one of the most influential players in cartilage destruction in OA [24]. BMMSC-Exo injection significantly reduces IL-1β-mediated suppression of chondrocyte proliferation, and thereby inhibits the downregulation of anabolic markers COL2A1 and ACAN, as well as blocks the overexpression of catabolic markers MMP13 and ADAMTS5. This treatment reduced both cartilage damage and knee pain in rats with OA [25]. In addition, exosomes derived from human embryonic stem cell-induced mesenchymal stem cells (ESC-MSCs) have shown therapeutic promise for the treatment of OA. In the presence of IL-1β, exosomes from ESC-MSCs were able to modulate chondrocytes by increasing the expression of the extracellular matrix protein collagen type II and decreasing the expression of the matrix degradation enzyme ADAMTS5. This effectively prevented cartilage destruction in the progression of OA [26]. A study has shown that high expression of miR-135b in TGF-β1-stimulated BMMSC-Exos (BMMSC-Exos^TGF-β1^) reduces pro-inflammatory factors and alleviates cartilage damage in OA. This is likely achieved by targeting MAPK6, which in turn boosts the M2 polarization of synovial macrophages [27] (Figure 1). Another study demonstrated that IL-1β-induced damage to synovial fibroblasts might be mitigated by exosomal miR-26a-5p, from human bone MSCs (hBMSCs), by hindering prostaglandin-endoperoxide synthase 2 (PTGS2) [28] (Figure 1).

WNT5A has a dual role, functioning in both chondrogenic differentiation and cartilage degradation, and it is involved in the destruction and degradation of cartilage in the formation of OA [29]. Exosomal miR-92a-3p, released from MSCs, precluded cartilage destruction in an OA mouse model by downregulating WNT5A expression [30] (Figure 1). Furthermore, WNT5A and WNT5B were substantially expressed in synovial MSC exosomes (SMSC-Exos). WNT5A and WNT5B carried by exosomes stimulated Yes-associated protein (YAP) activation via an alternative Wnt signaling pathway, which contributed to the enhancement of chondrocyte proliferation and migration. Overexpression of miR-140-5p could block the side effect of YAP by suppressing *RalA*, which rescued SOX9, aggrecan, and collagen type II expression and restored extracellular matrix (ECM) secretion [31] (Figure 1).

In the cartilage tissues of patients with traumatic OA, increased E74-like factor 3 (ELF3) expression and decreased miR-136-5p expression were found. Similar to the aforementioned miR-140-5p, exosomal miR-136-5p derived from BMMSCs was also shown to promote chondrocyte proliferation, migration, and ECM secretion by downregulating ELF3 expression, lessening MMP-1 levels, and intensifying collagen II, aggrecan, and SOX9 levels, thereby inhibiting cartilage degradation [32] (Figure 1). Furthermore, miR-155-5p-overexpressing SMSC-Exos might stimulate chondrocyte proliferation and migration while decreasing apoptosis, as well as increasing ECM secretion by targeting Runx2 to prevent OA [33] (Figure 1). Similarly, exosomal lncRNA-KLF3-AS1 generated from MSCs was reported to reduce chondrocyte apoptosis and stimulate chondrocyte proliferation, via sponging miR-206 to enhance G-protein-coupled receptor kinase interacting protein-1 (GIT1) expression, which attenuates IL-1β-induced chondrocyte injury [34] (Figure 1). Another study explored the effect of infrapatellar fat pad (IPFP) MSC-derived exosomes (MSC^IPFP^-Exos) in OA. MiR-100-5p-abundant MSC^IPFP^-Exos prevented damage to articular cartilage and lessened the severity of OA by hampering apoptosis of chondrocytes through the mTOR-autophagy signaling pathway [35] (Figure 1).

### 2.2. Wound Healing

Wound healing is a dynamic physiological process that may be roughly broken down into four stages: hemostasis, inflammation, proliferation, and remodeling. In this complicated process, numerous aspects, including angiogenesis, proliferation and differentiation of fibroblasts, and immunomodulation, play their roles in different stages of wound healing [36]. The effect of exosomes on the healing of wounds has long been a subject of significant research and clinical interest. Studies show that extracellular signal-regulated kinase, STAT-3, and protein kinase B are just a few of the pathways that exosomes can target. These pathways are essential for aiding in and speeding up wound healing via upstream effectors, such as insulin-like growth factor-1, hepatocyte growth factor, nerve growth factor, and stromal cell-derived factor. Low quantities of TGF-β have been discovered in MSC-Exos in the umbilical cord, and, when exosomes are further laden with TGF-β cargo, they promote the remodeling of the matrix and vascularization [37].

In this section, we will summarize the therapeutic applications of exosomes and the current understanding of their underlying mechanisms.

#### 2.2.1. Stimulation of Angiogenesis

Neovascularization is a paramount process in wound healing and tissue regeneration (Figure 2). HiPSC-MSC-Exos were first found to exhibit the ability to stimulate angiogenesis in human umbilical vein endothelial cells [38] (Figure 2). Further research indicates plausible mechanisms behind this finding. UCMSC-derived exosomes (UCMSC-Exos) transmitted angiopoietin-2 (Ang-2) to human umbilical vein endothelial cells [39], thus enhancing their proliferative, migratory, and tube-forming capabilities. Additionally, another study reported that UCMSC-Exos also transmitted miR-21-3p to HUVEC, which increased angiogenesis and proliferation by inhibiting phosphatase and tensin homolog (PTEN), and sprouty homolog 1 (SPRY1) [40] (Figure 2). Exosomes derived from ADSCs-Exos can also trigger the proliferation and migration of vascular endothelial cells, thus promoting angiogenesis. In another study, it was found that transferring miR-125a from ADSC-Exos to endothelial cells increased angiogenesis by blocking the effect of delta-like 4 (DLL4), an inhibitor of angiogenesis [41] (Figure 2).

Another mechanism that has been often addressed in MSC-Exos is the Wnt/β-catenin signaling pathway, which plays a pivotal role in the proliferative phase of the wound healing process [42]. The transfer of Wnt4 from UCMSC-Exos to endothelial cells activated the Wnt/β-catenin pathway and increased the number of angiogenesis-related factors, including proliferating cell nuclear antigen (PCNA), cyclin D3, and N-cadherin, to promote angiogenesis for the healing of second-degree burn injuries [43]. In the rat skin burn model, UCMSC-Exos containing Wnt4 stimulated the Wnt/β-catenin pathway and the AKT pathway to increase angiogenesis and decrease heat stress-induced apoptosis, respectively [44]. ADSC-Exos hindered the H_2_O_2_-induced apoptosis of HaCaT cells via stimulating the Wnt/β-catenin signaling pathway, which may be advantageous for cutaneous wound healing [45].

The angiogenic efficacy of MSC-Exos has also been applied to diabetic wound healing, where hyperglycemia impairs angiogenesis and impedes the wound healing process [46]. ADSC-Exos that overexpressed nuclear factor-E2-related factor 2 (Nrf2), a transcription factor that favors anti-oxidative properties, not only promoted proliferation of endothelial progenitor cells and angiopoiesis, but also increased granulation tissue, growth factor levels, and vascularization in diabetic mouse wounds while decreasing inflammatory and oxidative stress-related proteins [47].

Recent investigations have found that exosomes derived from preconditioned MSC are more therapeutically efficacious than those obtained from MSC grown under standard conditions. By transferring miR-126, which downregulated PTEN and activated the PI3K/AKT signaling pathway [48], BMMSC-Exos with deferoxamine, a hypoxia-inducing agent, strengthened the proliferative, migratory, and proangiogenic capacity of human umbilical vein endothelial cell and diabetic wound healing.

#### 2.2.2. Proliferation and Collagen Synthesis of Fibroblast

To establish an intact epidermal barrier, fibroblasts proliferate and produce ECM, which includes fibronectin, type 1 collagen, and type 3 collagen. Accumulating evidence suggests that MSC-Exos may be absorbed by fibroblasts and thereby stimulate collagen production (Figure 3). HiPSC-MSC-Exos increased the production, migration, and secretion of type 1 collagen, type 3 collagen, and elastin by human dermal fibroblasts (DF) [38]. ADSC-Exos transferred to DF enhanced collagen-related proteins, including type 1 collagen, type 3 collagen, MMP1, bFGF, and TGF-β1, and activated PI3K/Akt signaling to maximize collagen deposition [49] (Figure 3). In addition, exosomes produced from fetal dermal MSC-derived exosomes (FDMSC-Exos) augmented the secretion, migration, and proliferation of adult DF via activation of the Notch signaling pathway [50] (Figure 3).

#### 2.2.3. Inhibition of Scar Formation

In the process of wound healing and tissue repair, the excessive accumulation of myofibroblasts and deposition of collagen results in aberrant scar formation. By controlling fibroblast transition and ECM remodeling, administrations of stem cell-generated exosomes have the potential to limit scar formation, and even accomplish scar eradication, striking a balance between therapeutic effects and aesthetic concerns (Figure 4) [51].

In the wound healing microenvironment, macrophages secrete TGF-β1, which initiates the differentiation of fibroblasts into myofibroblasts. Disrupting the TGF-β/Smad pathway to prevent TGF-β1-induced differentiation of fibroblasts into myofibroblasts is a straightforward strategy for minimizing fibrosis and scarring under these circumstances [52] (Figure 4). In this context, evidence suggested that UCMSC-Exos, enriched in miR-21, miR-23a, miR-125b, and miR-145, inhibited TGF-β/Smad2 and myofibroblast formation [53] (Figure 4).

#### 2.2.4. Extracellular Matrix Remodeling

Accumulating evidence has suggested that exosomes are able to remodel and adjust the protein composition of the ECM, modifying the ECM into a more preferable condition throughout the process of wound healing [54].

The ECM is composed of a variety of proteins, the most prevalent of which is collagen. Collagen possesses dual roles during different stages of wound healing. During the early phase of wound healing, stimulation of collagen synthesis is essential for restoration of wound strength, but excessive collagen during the late phase of wound healing results in scar formation [55]. It has been reported that, aside from promoting the formation of myofibroblasts, the aforementioned TGF-β/Smad pathway is also involved in the regulation of collagen synthesis. The upregulation of the TGF-β/Smad pathway intensifies the expression of the *COLI2* gene during the early phase of wound healing but decreases collagen I deposition during the late phase [53]. Through the targeting of this pathway, UCMSC-Exos promote favorable ECM composition throughout the healing process [56] (Figure 4).

Zhang et al. demonstrated that UCMSC-Exos transport 14-3-3ζ proteins and diminishes collagen deposition. 14-3-3ζ proteins facilitate the binding between YAP and phosphorylation-large tumor suppressor kinase, resulting in YAP phosphorylation and the blockage of Wnt/β-catenin signaling, which causes skin fibrosis during the remodeling phase of healing [10] (Figure 4).

Matrix metalloproteinases (MMPs) are components of the ECM that degrade excessive collagen. It has been suggested that ADSC-Exos trigger the ERK/MAPK pathway and augment the expression of MMP-3 [57] (Figure 4). Furthermore, Yang et al. revealed that miR-21 in ADSC-Exos regulates the PI3K/AKT pathway and elevates MMP-9 production, which accelerates HaCaT keratinocyte migration and proliferation [51,58] (Figure 4).

### 2.3. Female Infertility

Impaired fertility is manifested as a declining pregnancy rate and poor pregnancy outcomes [59]. By definition, infertility is the failure to establish a pregnancy within 12 months of regular sexual intercourse [60]. Female reproductive disorders pose great threats to women’s health and eventually contribute to infertility [61]. Stem cell-derived exosomes used in cell-free treatment have been shown to enhance the ovarian and uterine environments through their capacity for regeneration, which could potentially reverse female infertility.

#### 2.3.1. Promoting Follicular Development

Several reproductive diseases are associated with disrupted follicle development. While ovarian aging is linked to a drop in the number and quality of oocytes, premature ovarian insufficiency, also known as premature ovarian failure, is a disease characterized by senescence of the ovaries in women under the age of 40 [60]. The resulting follicular malfunction often manifests clinically as amenorrhea, estrogen deficiency, and hypergonadotropism [62]. In addition, polycystic ovary syndrome (PCOS) is another common disease with ovulatory dysfunction among reproductive-aged females [63].

The successful maturation of oocytes, which is pivotal for ovulation and pregnancy, is intimately connected with hormone regulation, granulosa function, and the ovarian environment [64]. Since the number of oocytes defines the reserve of primordial follicles [65], restoring the impaired follicular development is the key to improving ovulatory failure caused by ovarian reproductive disorders. Premature ovarian insufficiency is the most frequently adopted model for this line of research. In the vast majority of studies, anti-apoptosis and adjustment of sex hormone levels are hypothesized to be the underlying mechanisms of exosome therapy.

Several researchers have delved into the molecular pathway of how stem cell-derived exosomes exert their ability to inhibit apoptosis through their cargo (Figure 5). Human ADSC-Exos were found to repress apoptosis of human granulosa cells by regulating the SMAD pathway, which elevates mRNA and protein expression of SMAD2, SMAD3, and SMAD5 to a normal level [66] (Figure 5). Furthermore, exosomes derived from ADSC transfected with miR-323-3p eliminate the pro-apoptotic action of programmed cell death protein 4 (PDCD4) in PCOS cumulus cells [63] (Figure 5). Exosomes derived from amniotic fluid stem cells (AFSC-Exo) deliver miR-10a and reduce follicular atresia by targeting *Bim* and downregulating the apoptotic factor *Casp9* [67] (Figure 5). In parallel, exosomes collected from BMMSC-Exos, carrying miR-644-5p, impede apoptosis of ovarian GCs through targeting *p53* [68] (Figure 5), the upstream of *Bim*. Meanwhile, BMMSC-Exos are capable of activating the PI3K/AKT pathway in granulosa cells by suppressing PTEN, an inhibitor of this signaling pathway [69] (Figure 5). On the other hand, UCMSC-Exos highly express miR-17-5p, inhibiting sirtuin 7 (SIRT7) and its downstream genes *PARP1*, *γH2AX*, and *XRCC6*. This results in a decreased accumulation of reactive oxygen species (ROS) in granulosa cells, which has an anti-apoptotic effect [70] (Figure 5).

Regarding regulation of sex hormone levels, one study revealed that exosomal miR-21 is involved in enhancing estrogen secretion in hUCMSCs (Figure 5). Further effort was made to clarify the molecular mechanism underlying this observation. Exosomal miR-21 was found to downregulate large tumor suppressor 1 (LATS1), which belongs to the Hippo pathway, thereby mitigating phosphorylation of lysyl oxidase-like 2 (LOXL2) and YAP. In this way, YAP protein could bind to the promotor of the steroidogenic acute regulatory (*StAR*) gene, thereby elevating *StAR* expression and promoting estrogen secretion in ovarian granulosa cells [71] (Figure 5).

Aside from acting on granulosa cells, exosomes also have an effect on oocytes (Figure 5). For instance, the mechanism of miR-320a, from human amniotic MSC-derived exosome (AMSC-Exo), inhibits oocyte apoptosis by suppressing SIRT4 signaling and reducing ROS levels [72] (Figure 5), an effect which is similar to that of miR-17-5p from BMMSC-Exos. In the model of age-related fertile retardation, UCMSC-Exos carries miR-146a-5p, which triggers the PI3K/mTOR pathway, improving oocyte quality and accelerating follicular development [65] (Figure 5). By means of promoting follicular development and obliterating atresia, stem cell-derived exosomes restore ovarian function and negatively feedback on follicle-stimulating hormone (FSH) levels.

In addition to the above-mentioned mechanisms, several studies indicate that exosome therapy can adjust ovarian stroma into a more favorable microenvironment. Zhang et al. showed that exosomes from menstrual blood-derived stromal cells not only stimulated the expression of early follicle markers Deleted in Azoospermia-like (DAZL) and Forkhead Box L2 (FOXL2), but also regulated the composition of the ECM. Enhanced expression of collagen IV, FN1, and laminin in the ovarian stroma was noted; the latter two were also observed in granulosa cells, and are involved in follicular growth [64] (Figure 5).

#### 2.3.2. Regeneration of Damaged Endometrium

Being the implant sites of blastocytes, endometrium undoubtedly plays a crucial role in female pregnancy. Asherman’s syndrome, also called intrauterine adhesion (IUA), is characterized by poor glandular epithelial growth and vascular development from scarring, and fibrosis primarily caused by artificial trauma, repeated endometritis, and infection [59,73,74]. Endometrial fibrosis causes a loss of regeneration competence in functional endometrium, which eventually leads to secondary infertility, the most common type of infertility worldwide [60,75].

Recent research has found that the administration of stem cell-derived exosomes has similar effects to PRP in endometrial fibrosis and AS models, including promoting cellular proliferation, differentiation, functional recovery, angiogenesis, anti-inflammation, and anti-fibrosis. Saribas et al. utilized uterus-derived MSCs-exosomes (uMSC-Exo) in AS models and found increased proliferation, vascularization, and reduced fibrosis in uterine tissue. The effect was confirmed by significantly elevated proliferating cell nuclear antigen (PCNA), VEGFR-1, MMP-2, MMP-9, and declined TIMP-2 [76] (Figure 6). Concerning specific molecular interactions between exosomes and endometrial cells, Shao et al. established a negative regulatory relationship between lncRNAs and miRNAs. They elucidated that ADSC-Exos transfer lncRNA-MIAT by targeting miR-150-5p in endometrial epithelial cells, and alleviate fibrosis of endometrial tissue with a higher expression of cytokeratin 19 and a lower expression of fibrosis markers α-SMA and TGFβR1 [75] (Figure 6).

### 2.4. Cardiac Recovery

The protective effect and recovery potential of exosomes derived from stem cells have been sought for use in the prevention or treatment of cardiovascular diseases (CVDs). Via the action of shuttling and releasing cardioprotective molecules or miRNAs, stem cell-derived exosomes displayed considerable therapeutic capacities for myocardial repair, cardiac function recovery, and neoangiogenesis, providing alternative perspectives for clinical therapies for CVD patients in the future.

#### 2.4.1. Treatment of Cardiovascular Diseases

In recent years, accumulating research has verified the treatment of CVDs with secreted exosomes instead of their stem cell resources [77] (Figure 7). A study by Shao et al. compared the effectiveness of BMMSC and BMMSCs-Exos in cardiac repair. It was found that the effects of BMMSCs-Exos, in reducing inflammation, inhibiting fibrosis, and restoring cardiac function, were significantly superior to those of BMMSCs. Further miRNA sequencing found a similar expression profile between BMMSC and BMMSCs-Exos, with a higher expression of several cardiac-protective miRNAs, such as miR-29 and miR-24, in BMMSCs-Exos [1] (Figure 7). Results from another study suggested that the miR-24 contained in MSC-Exos could activate the PI3K/AKT pathway by inhibiting PTEN, which leads to an amelioration of cardiomyocyte apoptosis in hypoxic conditions [78] (Figure 7). Moreover, treatment with exosomes from ADSCs increased miR-221/222 expression, and thereby downregulated the expression of the apoptosis-related protein PUMA and hypertrophy-related protein ETS-1 in cardiomyocytes of I/R mice [79] (Figure 7). Studies indicate that miR-22-enriched exosomes produced by MSCs decreased cardiomyocyte apoptosis in ischemia circumstances. When miR-22 is delivered to fibroblasts under hypoxic circumstances, cardiomyocyte apoptosis and fibrosis are decreased, and post-MI cardiomyocyte angiogenicity is boosted. The pro-fibrotic methyl CpG binding protein 2 (Mecp2) is similarly downregulated by miR-22, which leads to anti-apoptotic and healing effects. Exosomal miR-132, which is produced from human perivascular pericytes and is transferred to ECs, also plays a role in the suppression of pro-fibrotic MECP 2. Bristol pericytes maintained heart function by reducing scarring and promoting the flow of blood and angiogenesis in a mouse model of MI, demonstrating the protective role of miR-132 in exosomes [80], particularly when TGF-β is present, which activates miR-132. Besides, p120RasGAP can be downregulated by miR-132, which upregulates the previously suppressed gene to enhance angiogenesis [81].

One study investigated the effect of MSC-Exos on an aging heart. It was discovered that UCMSC-Exos prevent aging-induced cardiac dysfunction by delivering lncRNA metastasis-associated lung adenocarcinoma transcript 1 (MALAT1), which downregulated the NF-κB/TNF-α pathway [82] (Figure 7).

#### 2.4.2. Recovery from Myocardial Infarction (MI)

Several studies have emphasized the potential of stem cell-derived exosomes in recovery from myocardial infarction (MI) (Figure 8).

In MI mice, BMMSC-Exos could shuttle miR-185 to improve cardiac function and protect cardiomyocytes from apoptosis via inhibition of the expression of suppressor of cytokine signaling 2 (SOCS2) [83] (Figure 8). A study by Khan et al. found that mouse embryonic stem cell-derived exosomes (mES-Exos) induce neo-vascularization, promote myocyte survival, and augment post-MI cardiac function. These effects are achieved through miR-294 delivery, which enhances the survival and proliferation of resident cardiac progenitor cells [84]. Moreover, transfection of exosomes from hMSCs with overexpressed lncRNA KLF3-AS1 exhibit the ability to reduce MI area and suppress MI progression. KLF3-AS1 in hMSC-derived exosomes hampered apoptosis and pyroptosis of cardiomyocytes by sponging miR-138-5p, and subsequently upregulating Sirt1 expression to attenuate MI progression [85] (Figure 8).

Importantly, several studies revealed that overexpression of HIF-1α in exosomes manifested therapeutic effects on MI. It is suggested that overexpression of HIF-1α in MSC-Exos is able to rescue the impaired angiogenesis of hypoxia-injured human umbilical vein endothelial cells in MI condition, which was mediated via VEGF and PDGF [86] (Figure 8). Furthermore, MSC-Exos with HIF-1α overexpression also exerts angiogenic and anti-apoptotic effects, by upregulating the expression of miR-221-3p in cardiomyocytes [87] (Figure 8).

### 2.5. Immunomodulation

The immune system plays a crucial role in directing tissue repair and influencing regenerative outcomes. Adult stem cell-derived exosomes have been shown to display immunomodulatory and anti-inflammatory regenerative effects in several diseases and tissue injuries [88].

#### 2.5.1. Macrophage Polarization

An accumulating amount of evidence has revealed that MSC-Exos direct macrophages from a pro-inflammatory phenotype (M1) toward an anti-inflammatory phenotype (M2) (Figure 9). M2 macrophages produce mediators and cytokines that aid in the reduction of inflammation, tissue remodeling, and wound healing. Several diseases can be ameliorated with M2 polarization [89]. Moreover, it has been reported that BMMSC-Exos transfer miR-223 into macrophages and promote macrophages to the M2 phenotype by diminishing the pknox1 protein level in macrophages, which in turn accelerates cutaneous wound healing [90] (Figure 9).

Toll-like receptor 4 (TLR4) is a receptor expressed on immune cells that responds to inflammatory responses and mediates inflammatory signal transduction. Inhibition of the TLR4 pathway has been associated with M2 polarization and exerted protective effect on various disease models [91] (Figure 9). Research by Ti et al. demonstrated that LPS pre-Exos possess the advantage of switching macrophages to an M2 state by shuttling let-7b, which directly downregulates TLR4 signaling and the subsequent NF-κB activity. This contributes to a reduction of inflammation and the promotion of tissue regeneration. Additionally, mouse BMMSC-Exos are reported to relieve and repair myocardial I/R injury through the regulation of TLR4 activity. The miR-182 delivered by BMMSC-Exos inhibits the expression of TLR4 within macrophages and induces M2 polarization [92] (Figure 9).

#### 2.5.2. Regulation of T and B Lymphocytes

Multiple studies have also demonstrated that stem cell-derived exosomes possess remarkable immunosuppressive capacity. Through regulating the proliferation and activation of T cells and the differentiation and maturation of B cells, stem cell-derived exosomes often attenuate inflammatory responses and promote a tolerogenic immune environment [93] (Figure 10).

The diverse mechanisms underlying the immunoregulatory potential of stem cell-derived exosomes in T cells have been explored. Stem cell-derived exosomes carry various active molecules, such as active CD73 protein, miR-125a-3p, miRNA-181a, indoleamine 2,3-dioxygenase (IDO) protein, and GM-CSF, which endow them with the ability to suppress T cell proliferation [94,95,96,97,98].

Studies have shown that MSC-Exos inhibit in vitro T cell proliferation via adenosinergic signaling. CD73, which is expressed in MSC-Exos and carries ATPase activity, catalyzes the production of adenosine 5′-monophosphate (AMP). The interaction between exosomes derived from CD73-MSC and CD39-positive T cells leads to the efficient production of adenosine and the subsequent suppression of immune responses [94] (Figure 10). Moreover, several recent studies have reported the possibility of utilizing exosomes as a potential treatment for acute graft-versus-host disease (aGVHD). A study on aGVHD in mice suggested that miR-125a-3p, delivered by human BMMSC-Exos, downregulates T cell proliferation and differentiation, and thereby prolongs the survival time of circulating naive T cells. This contributes to the amelioration of clinical GVHD symptoms and pathological damage [95] (Figure 10). Furthermore, as a potential therapeutic strategy for I/R injury, miRNA-181a delivered by MSC-Exos promotes Treg development and reduces the inflammatory response via inhibition of the c-Fos protein [96] (Figure 10). IDO has also been proposed as a promising immunomodulator for the inhibition of allograft rejection. The rat BMMSC-Exos overexpressing IDO1 increased the number of Tregs and decreased the number of CD8+ T-cells, which may promote immunotolerance and improve the survival of cardiac allografts [97] (Figure 10). On the other hand, expression of GM-CSF in exosomes from murine ESCs increases CD8+ T-cells but inhibits Tregs in tumor cells, which provides a novel concept of cancer prevention by cell-free exosome-based vaccination [98] (Figure 10).

Concerning the immunomodulatory effects of stem cell-derived exosomes on B cells, studies have indicated that, in inflammatory conditions, miR-155-5p in MSC-Exos directly reduce cell viability in activated B cells by downregulating the PI3K-AKT signaling pathway [99] (Figure 10). The immunosuppressive effect and mediators that control B cell activation, proliferation, and migration still warrant further investigation. The crosstalk between stem cell-derived exosomes and B cells can be viewed as a novel potential therapeutic target.

### 2.6. Neurological Disease

Neurologic complications are commonly regarded as irreversible impairments due to the limited potential for regeneration of the central nervous system. MSC-Exos have become novel neurological therapeutic approaches in light of their potential for neuronal regeneration, neuroprotection, and neuroinflammatory modulation [100].

#### 2.6.1. Stroke

Ischemic stroke, the most common type of stroke, results from a blockage of circulation and causes irreversible neuronal damage. It has been suggested that MSC-Exos increase the number of neuroblasts and endothelial cells in stroke rats, as well as stimulating neurite remodeling and synaptic plasticity [101] (Figure 11). Additionally, neural stem cell (NSC)-derived exosomes were also implicated in neuroprotection following stroke. In a study on post-stroke mice, NSC-Exo was found to protect astrocytes from oxygen–glucose deprivation injury and also reduce infarct volumes [102] (Figure 11).

Further investigations revealed possible mechanisms behind those effects. The transfer of miR-133b from MSC-Exos to astrocytes and neurons decreased gene expression of connective tissue growth factor and *RAS* homolog gene family member A, which benefits axonal plasticity, neurite remodeling, and functional recovery in stroke rats [103] (Figure 11). Additionally, miR-17-92 from MSC-Exos also enhances oligodendrogenesis, neurogenesis, and neurite remodeling in stroke rats via the inhibition of PTEN and subsequent activation of the PI3K/Akt/mTOR/GSK-3β signaling pathway [104] (Figure 11).

#### 2.6.2. Alzheimer’s Disease

Alzheimer’s disease (AD) is the most common cause of dementia. However, no effective therapy has been established to reverse the neurodegenerative progression. [105] The major characteristics of AD are the deposition of amyloid β peptides (Aβ) and the subsequent neuroinflammation [106], loss of synaptic transmission [107], and neuronal dysfunction [108]. The application of MSC-Exos was shown to alleviate cognitive impairment and promote neurogenesis in the subventricular zone in AD mice [109]. The transfer of miR-21 from hypoxia-preconditioned MSC-Exos to AD mice rescued cognitive and memorial impairment, and reduced Aβ levels in the brain [110]. Additionally, ADSC-Exo may also alleviate AD by hindering neuronal cell apoptosis and promoting neurite growth [111] (Figure 11). Moreover, in order to enhance the specificity of MSC-Exo to the brain, rabies viral glycoprotein (RVG), a central nervous system-specific peptide, was conjugated with MSC-Exos to treat AD-induced cognitive impairment [112]. A reduced Aβ deposition, astrocyte activation, and neuroinflammation was demonstrated by targeting MSC-Exos to the cortex and hippocampus, and a prominent improvement in learning and memory capacity was seen [112] (Figure 11).

In patients with AD, Aβ aggregates induce neuroinflammation and activate microglia [113]. Microglia can also be categorized into two different subtypes as macrophages: the pro-inflammatory “M1 microglia” and the anti-inflammatory “M2 microglia”. The M2 microglia are associated with higher levels of Aβ-degrading enzymes, such as insulin-degrading enzyme and neprilysin, in the brain [114]. UCMSC-Exos were found to alternatively activate M2 microglia and increase the level of insulin-degrading enzyme and neprilysin in vivo, which may alleviate the neuroinflammation caused by Aβ deposition and the related AD cognitive impairments [115] (Figure 11). On the other hand, the hypoxia-preconditioned MSC-Exos inhibited activation of astrocytes and microglia, and promoted the transformation of microglia into dendritic cells [110], which degrade and uptake Aβ to lower amyloid deposition (Figure 11).

### 2.7. Metabolic Regulation

Targeting the alterations in cellular metabolic pathways in various diseases is another emerging research field associated with stem cell-derived exosomes. In the treatment of type 2 diabetes mellitus (T2DM), improving hepatic glucose and lipid metabolism represents an important strategy. It has been shown that UCMSC-Exos not only increased insulin sensitivity in the T2DM model, but also improved glucose and lipid metabolism both in vitro and in vivo by promoting hepatic glycolysis, glycogen synthesis, and lipolysis and inhibiting gluconeogenesis. AMPK-dependent autophagy may be involved in these efficacies [116].

Macrophage polarization plays a crucial role in the treatment of sepsis-induced acute respiratory distress syndrome (ARDS) through the modulation of the excessive inflammatory response occurring in ARDS. It was found that intratracheal delivery of BMMSC-Exos may be an effective treatment option for ameliorating inflammation and lung injury by promoting M2 polarization in LPS-treated MH-S cells (a murine alveolar macrophage cell line) via HIF-1α inhibition and cellular glycolysis downregulation [117].

### 2.8. Cancer

The tumor microenvironment, consisting of the extracellular matrix, immunity cells, and stromal cells (e.g., fibroblasts, MSCs, pericytes) [118], plays a crucial role in tumorigenesis, metastasis, progression, and response to treatment [119]. In this context, exosomes, produced by cancer cells and tumor microenvironment-associated cells, are proven to transport signaling molecules that facilitate cell-to-cell communication in the tumor microenvironment by delivering proteins, nucleic acids, and lipids [120]. Accumulating evidence has demonstrated the role of MSC-Exos in tumor development, due to its ability to modulate the tumor microenvironment. Multiple lines of research have so far indicated that MSC-Exos are involved in a multitude of mechanisms that could lead to either the promotion or attenuation of cancer treatment resistance. It has been found that the MSC-Exos could enhance therapeutic sensitivity in several cancers, including lymphoma, breast cancer ovarian cancer, gastric cancer, and hepatocellular carcinoma [121]. Mechanistically, MSC-Exos hinder the development of these cancers through the inhibition of tumor proliferation and angiogenic pathways, and through the stimulation of apoptosis. Conversely, MSC-Exos can also promote proliferation and induce dormancy and chemoresistance in cancer cells, which contradicts their aforementioned roles. This discrepancy may be related to the difference in experimental conditions, the heterogeneity of MSCs adopted by different studies, the complexity of tumor microenvironments, and the diversity of malignancy origins [122].

Interestingly, exosomes from cancer cells could also create a “pre-metastatic” environment in distant organs that are prone to metastasis [123]. A study by Costa-Silva et al. suggested that pancreatic cancer cell-derived exosomes promote TGF-β signaling which activates hepatic stellate cells and ECM remodeling. This in turn induces the translocation of bone marrow-derived macrophages to the liver and provides a favorable environment for liver metastasis [124]. Another study reported that breast cancer cell-derived exosomes promoted bone metastasis by transferring miR-21 to osteoclasts, which is associated with the formation of a pre-metastatic environment [125].

On the other hand, the delivery capability of exosomes also makes them suitable carriers of anti-cancer drugs and RNA. One study showed that engineered miR-21 sponge-containing exosomes prevent cancer development by downregulating miR-21 in glioma cells, and thus promoting the expression of miR-21 target genes PDCD4 and RECK [126]. The presence of transmembrane and membrane-anchoring proteins is a significant advantage of exosomes as a drug carrier, as they facilitate endocytosis and markedly enhance their delivery capacity [127]. A study by Kim et al. found that the cellular uptake of paclitaxel in mouse lung cancer cell lines was significantly higher with macrophage-derived exosomes than with liposomes [128,129]. Additionally, exosomes can be used to initiate antitumor immune responses. A study showed that α-galactosylceramide/ovalbumin-loaded exosomes induced T cell response and decelerated tumor growth in melanoma mice [130]. Overall, exosomes from stem cells or cancer cells not only play an important role in cancer progression and signal transduction in tumor microenvironment, but also show their therapeutic potential as a carrier of anti-cancer therapies.

## 3. Strategies That Enhance the Therapeutic Effects of Exosomes

With an increasing number of investigations seeking to elucidate the mechanistic and therapeutic characteristics of exosomes, another line of research has appeared, to improve the isolation technique and optimize the biomaterial properties of exosomes.

### 3.1. Effective Isolation of Exosomes

Isolation of exosomes serves as one of the most fundamental aspects of exosome research. However, the development of a simple, high-yield, and discriminative extraction procedure remains challenging. Generally, the major technical conundrums that have hindered the advancement of exosome research are the simplification of the exosome extraction process, the improvement of extraction yield, and the purification of exosomes from other EVs [131]. Exosome samples prepared by current techniques inevitably contain various kinds of non-exosome vesicles. Among those vesicles, the existence of certain functional microvesicles jeopardizes the reliability and authenticity of exosome-based research [132].

The traditional methods of exosome isolation were based on the physicochemical properties of exosomes. Commonly adopted methods include ultracentrifugation, filtration, and polymer precipitation. In the past decades, various ultracentrifugation strategies have been the gold standard for isolating exosomes, and among these strategies, density-gradient centrifugation was widely used due to it yielding the purest exosome samples [133]. However, ultracentrifugation requires expensive equipment, and, importantly, it may cause mechanical or functional damage to the exosomes from high speed and prolonged centrifugation [134]. To enhance simplicity and efficiency, filtration techniques, including ultrafiltration, hydrostatic filtration, dialysis, and gel filtration (also known as size exclusion chromatography, or SEC), have gained increasing popularity. Furthermore, methods utilizing the change in exosome solubility and aggregation have been developed. These methods employ various hydrophilic polymers, including precipitation with polyethylene glycol, protamine, and sodium acetate, to induce precipitation of exosomes, and exhibit improved extraction yields with lower cost [135].

Recently, novel strategies such as immunoaffinity binding and microfluidic-based methods have been developed [136]. The former is functions by the specific binding of exosome surface markers to immobilized antibodies, and boosts the ability to provide pure exosome preparation with a limited volume of sample [133]. On the other hand, microfluidic-based methods are miniaturized extraction platforms, based on the above-mentioned mechanisms. This development allows the rapid isolation of exosomes with decreased reagents, cost, and time.

### 3.2. Scaffolds

Scaffolds are porous, 3-dimensional (3D) biomaterials that provide structural support and are widely utilized in the biomedical field for enhanced cell adhesion, ECM deposition, and molecular mobility. Recently, the design of scaffolds as stem cell or exosome carriers has been considered promising, as better therapeutic effects have been exhibited by exosomes derived from stem cells cultured in 3D scaffolds than in 2D conventional culture conditions [137]. Compared with random structures, scaffolds possess superior cell recruitment ability and promote the stable release of exosomes, which allows for enhanced absorption of exosomes by target cells [138].

Scaffolds made of different materials have been increasingly designed and reported. The ideal material for scaffold construction possesses the following properties: biocompatible, non-inflammatory, highly porous, biodegradable, and relatively inexpensive. An alginate scaffold that slowly releases UCMSC-Exos has been used to treat nerve injury-induced pain in a rat model. This bioresorbable biomaterial may provide a protective environment for UCMSC-Exos in vivo. With this scaffold, a better antinociceptive effect was shown compared to the group without the scaffold [139]. Moreover, a collagen scaffold laden with UCMSC-Exos has shown potent capacity for improving endometrium regeneration and collagen remodeling, and thereby acting as a potential treatment strategy for restoring fertility [140]. Additionally, as a novel therapeutic method to restore large-sized bone defects, polydopamine-coating PLGA (PLGA/pDA) scaffolds were reported to be able to promote the consistent release of ADSC-Exos and support bone regeneration via facilitating human BMMSC migration and proliferation [141].

Lately, 3D printed scaffolds have been adopted to improve bone and cartilage regeneration in animal models [138,142]. A 3D printed titanium alloy scaffold (Ti-scaffold), filled with osteogenic exosomes from human MSCs, was shown to facilitate bone regeneration [142]. Another 3D printed scaffold fabricated with decellularized cartilage ECM, MSC-Exos, and gelatin methacrylate (GelMA) hydrogel (ECM/GelMA/exosome scaffolds) was suggested as a promising strategy for early OA therapy. It was shown that 3D printed ECM/GelMA/exosome scaffolds possess radially oriented channels that induce chondrocyte migration and support cartilage regeneration [138].

### 3.3. Hydrogels

Hydrogels are hydrophilic polymers characterized by a highly porous structure and biodegradability. Given that the conventional injection or topical administration of exosomes is prone to being rapidly eliminated and leading to treatment difficulties, hydrogels, which share similar characteristics with the native ECM, have been widely used as a sustained carrier of exosomes [143]. Through placing hydrogels at or near the wound areas, exosomes can be incorporated with hydrogels, which promotes better therapeutic effects and tissue repair [144]. Hydrogels do not have significant bioactivities by themselves, but employing hydrogels that have been loaded with EVs can increase their stability, and assist EVs in reaching the injured tissue for an extended in situ release [145]. Many investigations have shown that exosome-loaded hydrogels have a strong potential for use in the regeneration or repair of tissues [146]. Almost all forms of tissue injury, including those to the skin, heart, central nervous system, bone, reproductive systems, and cartilage, may be effectively treated with it.

Another aspect influencing hydrogel loading is the interactions between oppositely charged EVs’ phospholipid membranes and the glycocalyx-charged residues, which react with the charge on biocompatible material in a repellent or attractive way [147]. Cationic delivery techniques, for instance, can be used for chitosan-based hydrogels. Cationic delivery methods, such as chitosan-based hydrogels, can bind exosomes via electrostatic forces and extend the loading time [148]. Furthermore, surface-expressed adhesion molecules, such as integrins, enable exosomes to attach to components of the ECM matrix; this interactivity might be exploited to modulate the release of exosomes from hydrogels [147].

Since unconjugated exosomes may present difficulties when administered directly, due to fast in vivo eradication by innate immunity, certain ligands have been conjugated to the exterior of exosomes in an attempt to change cellular interactions with exosomes and, perhaps, exosome biodistribution. In this case, a hydrogel could provide additional benefits in hindering rapid clearance of EVs in tissue, with its biocompatible advantages. [149].

As for the applications of hydrogels, studies have shown that the incorporation of chitosan hydrogel and human placenta-derived MSC-Exos is beneficial for tissue regeneration after ischemic injury, increasing the angiogenesis of ischemic tissue by improving retention of exosomes in a rat model [150]. Furthermore, in the treatment of spinal cord injury, human MSC-derived exosomes fixed in a peptide-modified adhesive hydrogel (Exo-pGel) promote the 3D adhesion and retention of human MSC-Exos, facilitating nerve regeneration, nerve repair, and motor functions in rats [151]. In skin regenerative therapies, an alginate hydrogel, loaded with ADSC-Exos, can considerably promote collagen synthesis, wound closure, and vessel formation in the wound site [152]. Moreover, it has been indicated that a genipin crosslinked hydrogel—a thermo- and pH-sensitive hydrogel, synthesized with poloxamer 407 and carboxymethyl chitosan, which encapsulates hUCMSCs-Exos—improves wound closure and collagen deposition, enhances re-epithelialization rates, and reduces inflammatory response. To provide a promising strategy for promoting diabetic wound healing, SIS/MBG@Exos hydrogel scaffolds, constructed with decellularized small intestinal submucosa (SIS), mesoporous bioactive glass (MBG), and MSC-Exos, have been suggested [153]. Studies have demonstrated that SIS/MBG@Exos hydrogel with ADSC-Exos accelerates angiogenesis and promotes collagen deposition, which could serve as a novel strategy for improving wound healing [154,155].

There is a future for hydrogels; however, apart from deciding the optimal material for hydrogels, other challenges, such as the potential toxicity from unreacted cross-linkers in hydrogel production, or the clogging of needles due to temperature or pH sensitivity, also warrant consideration [143].

### 3.4. Other Modalities

In addition to scaffolds and hydrogels, there are several other novel approaches for elevating the extraction yield, retention, and delivery of exosomes with a view to subsequently improving their therapeutic effects [156,157,158]. A polysaccharide-based dressing (FEP), fabricated with Pluronic F127 grafting polyethylenimine and aldehyde pullulan, possesses multifunctional features, including antibacterial activity and the capability of fast hemostasis, self-healing, and UV-shielding. This dressing plays an important role in maintaining function and sustaining the release of exosomes. In a diabetic wound healing model, ADSC-Exos loaded with FEP@exo dressing showed enhanced capability in stimulating angiogenesis, skin regeneration, granulation tissue formation, and collagen formation [156]. The introduction of nanoparticles has also become a promising approach, given that it permits the elevation of the exosome extraction yield. Studies have revealed that the introduction of positively charged nanoparticles into MSCs induces the generation and release of exosomes by stimulating multiple vesicle bodies formation. This finding provides a potential solution for future exosome-based therapies through the development of nanoparticle-based technology [157].

## 4. Challenges and Barriers in Clinical Application of Exosomes

Exosome therapy has its challenges [159], despite the fact that it represent a potential treatment for various diseases and tissue regeneration. Stem cell-derived exosomes are the type of exosome treatment that is most frequently considered. Exosomes derived from stem cells are most desirable for their intrinsic therapeutic properties, which are identical to those of their original cell. These exosomes are useful and practical for treating a range of illnesses, in this regard. However, the issue that quickly surfaces is that naturally occurring exosomes transport a vast array of biomolecules, with unforeseeable consequences [160]. As an illustration, miRNA, a typical exosome cargo, enables a single miRNA to control several genes, and may have a variety of off-target impacts [161]. The reliability of native exosome treatments is further complicated by the fact that cells produce exosomes with varying payloads in response to changes in ambient factors. Thus, to guarantee exosome uniformity, quality control procedures are required before prescribing stem cell-derived native exosome therapy [162]. Every step of the manufacturing process—including cell lines, exosome separation, characterization, alteration, and storage—is impacted by this.

In diverse disorders, exosomes have been identified from different bodily fluids, including breast milk, bile, blood plasma, gastric juice, and saliva. Exosomal output obviously varies with sex, illness severity, age, and lifestyle choices, therefore each kind of specimen requires a different optimal technique. Additionally, it is critical to understand how to preserve exosomes throughout the extraction phase, due to the various levels of deterioration within those samples [163]. Size exclusion chromatography, density gradient, immunoaffinity, microfluidics, ultracentrifugation, and many more techniques are employed to extract exosomes. Nonetheless, these techniques offer both benefits and drawbacks. It will be crucial to choose an effective isolation technique carefully for the specific sample. Additionally, exosome preservation is essential, since they can lose their properties and deteriorate amid freeze–thaw cycles. Their viability is also severely impacted by the treated anticoagulants when undergoing isolation, such as citrate, ethylenediamine tetraacetic acid, or heparin [164].

The next challenge is to define the exosomes, and this is a crucial step in determining their dose for therapeutic application, which comes after meticulous isolation and preservation [132]. Exosome characterization procedures need to be harmonized throughout general practice. All of these methods must also be robust and flexible, in order to enable the generation of exosomes in a quantity that is clinically meaningful. Eventually, before exosome-based therapies become widely used, they must be ameliorated, since such a complex system necessitates the coordination of resources and financial investment [165].

## 5. Conclusions

Stem cell-derived exosomes exhibit their therapeutic effects in various diseases through contained or secreted biological factors. Stem cell-derived exosomes, especially MSC-Exos, have been widely investigated in several conditions recently. They are a promising clinical treatment for conditions such as OA, wound healing, female infertility, cardiac dysfunction, and neurological diseases, and are involved in immunomodulation and metabolic regulation to maintain homeostasis in aged or injured tissues. Illuminating the underlying mechanisms and signaling pathways will be helpful for their development in clinical applications. Although exosomes possess advantages in treatment, they still face the challenges of achieving high-yield isolation and avoiding rapid elimination. Therefore, we further discuss the current approaches for effective exosome isolation, and the selection and special materials for improving the preservation of exosomes. The utilization of cell-free treatments in the future as an effective method of therapy has a great deal of promise.

## Figures and Tables

**Figure 1 cells-12-01018-f001:**
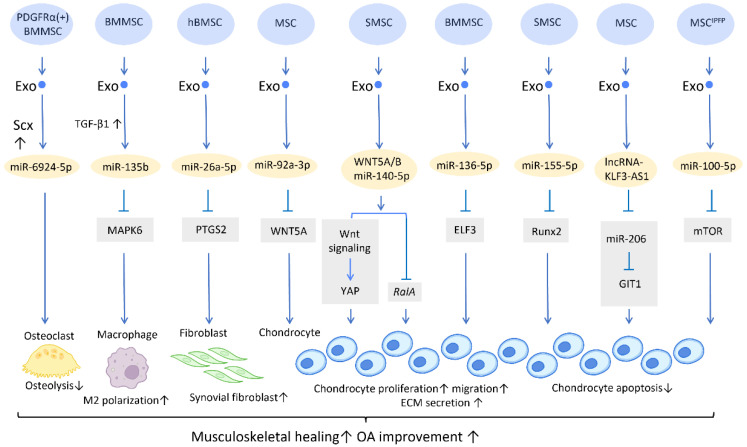
Molecular mechanisms of stem cell-derived exosomes in sport injuries and osteoarthritis. Stem cell-derived exosomes promote tissue healing, M2 polarization, fibroblast activation, chondrocyte proliferation and migration, and decreases osteolysis and chondrocyte apoptosis via transferring of diverse molecular components. An overall regenerative effect was demonstrated, which manifested as improved musculoskeletal healing and osteoarthritis recovery. Abbreviation: Scx: scleraxis; PTGS2: prostaglandin-endoperoxide synthase 2; YAP: Yes-associated protein; ELF3: E74-like factor 3; GIT1: G-protein-coupled receptor kinase interacting protein-1; hBMSC: human bone MSC; SMSC: synovial MSC; MSC^IPFP^: infrapatellar fat pad. ↑ represents upregulation; ↓ represents downregulation.

**Figure 2 cells-12-01018-f002:**
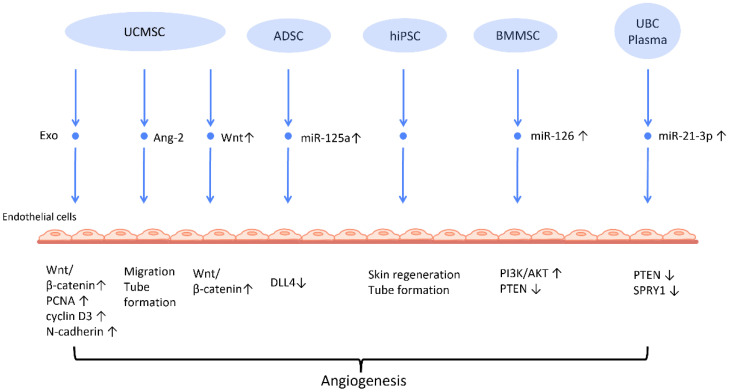
Molecular mechanisms of stem cell-derived exosomes in stimulation of angiogenesis. Stem cell-derived exosomes facilitate neovascularization by transmitting various angiogenic factors or miRNA to endothelial cells, leading to increased proliferation and migration of endothelial cells. Abbreviation: Exo: exosome; MSC: mesenchymal stem cell; UCMSC: human umbilical cord MSC; PCNA: proliferating cell nuclear antigen; ADSC: adipose-derived stem cells; DLL4: delta-like 4; hiPSC-MSC: human induced pluripotent stem cell-derived mesenchymal stem cells; BMSC: bone marrow-derived MSCs; Ang-2: angiopoietin-2; UBC: umbilical cord blood; PTEN: phosphatase and tensin homolog; SPRY1: sprouty homolog 1. ↑ represents upregulation; ↓ represents downregulation.

**Figure 3 cells-12-01018-f003:**
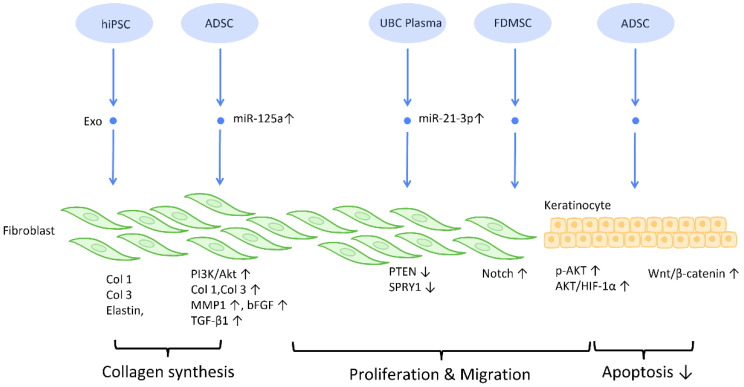
Molecular mechanisms of stem cell-derived exosomes in regulation of fibroblast and keratinocyte. Stem cell-derived exosomes enhance the proliferation and collagen synthesis of fibroblasts, thereby improving the healing process. ADSCs are also shown to accelerate keratinocyte migration and proliferation, which contributes to augmented healing activity. Abbreviation: Exo: exosome; Col: collagen; MSC: mesenchymal stem cell; ADSC: adipose-derived stem cell; miR: micro-RNA; TGFβ: transforming growth factor-β; PI3K/Akt: phosphatidyl-inositol 3-kinase/serine-threonine kinase; MMP: matrix metalloproteinase. ↑ represents upregulation; ↓ represents downregulation.

**Figure 4 cells-12-01018-f004:**
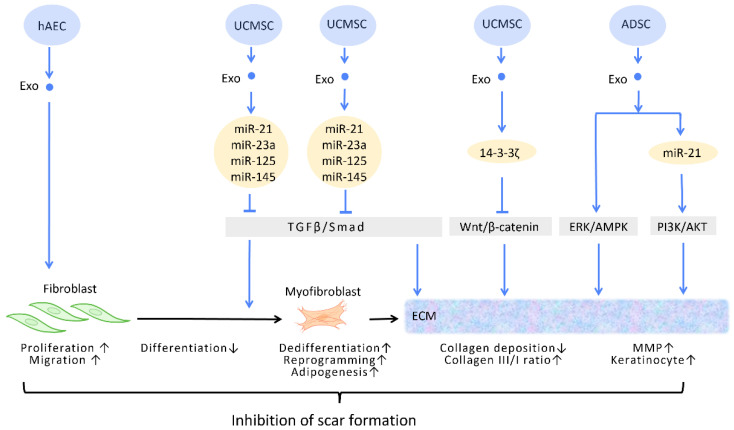
Molecular mechanisms of stem cell-derived exosomes in inhibition of scar formation and promotion of ECM remodeling. In the process of tissue repair, excessive accumulation of myofibroblasts and deposition of collagen result in aberrant scar formation. By controlling fibroblast transition and ECM remodeling, administrations of stem cell-generated exosomes have the potential to limit scar formation. Abbreviation: ECM: extracellular matrix; ERK: extracellular signal-regulated kinase; MAPK: mitogen-activated protein kinase; PI3K/Akt: phosphatidyl-inositol 3-kinase/serine-threonine kinase; MMP: matrix metalloproteinase. ↑ represents upregulation; ↓ represents downregulation.

**Figure 5 cells-12-01018-f005:**
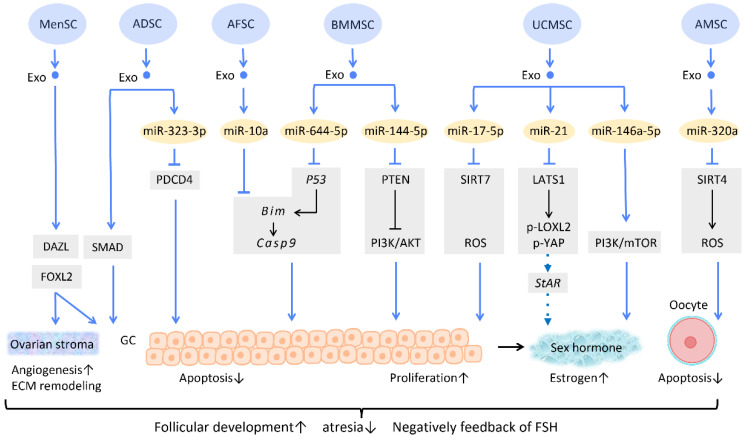
Molecular mechanisms of stem cell-derived exosomes in follicular development. Follicular malfunction is the critical reason of ovarian reproductive disorders. With the help of stem cell-derived exosomes, follicular growth and sex hormone levels could be restored, and ovarian microenvironment could be rejuvenated. Abbreviation: Exo: exosome; SC: stem cell/stromal cell; MSC: mesenchymal stem cell; MenSC: menstrual blood-derived SC; ADSC: adipose-derived stem cell; AFSC: amniotic fluid stem cell-derived; BMMSC: bone marrow-derived MSC; UCMSC: umbilical cord MSC; AMSC: amniotic MSC; miR: micro-RNA; DAZL: deleted in azoospermia like; FOXL2: forkhead box L2; PDCD4: programmed cell death protein 4; PI3K/Akt: phosphatidyl-inositol 3-kinase/serine-threonine kinase; SIRT: sirtuin; ROS: reactive oxygen species; LATS1: large tumor suppressor 1; LOXL2: lysyl oxidase-like 2; YAP: Yes-associated protein; StAR: steroidogenic acute regulatory gene; GC: granulosa cell; ECM: extracellular matrix. ↑ represents upregulation; ↓ represents downregulation.

**Figure 6 cells-12-01018-f006:**
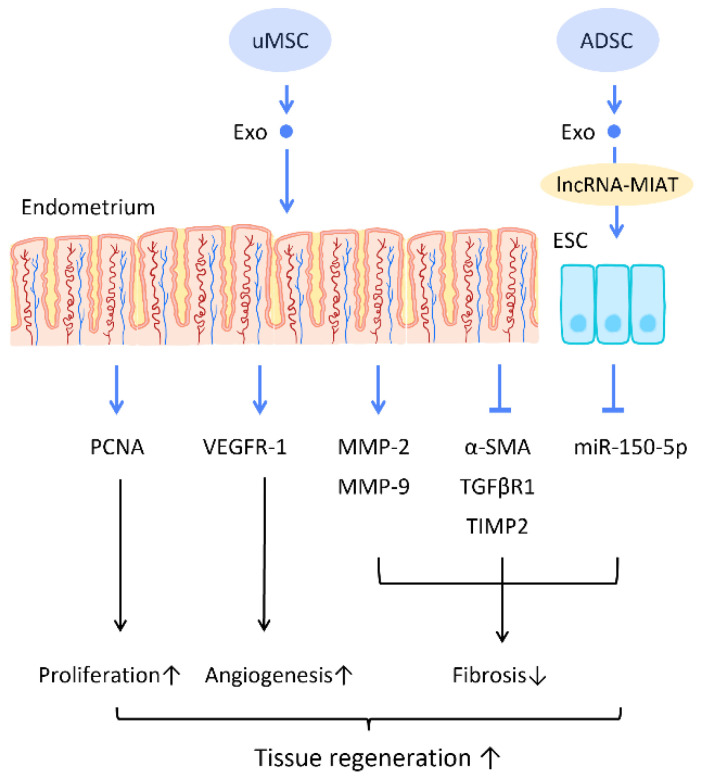
Molecular mechanisms of stem cell-derived exosomes regeneration of damaged endometrium. The administration of stem cell-derived exosomes is beneficial in female pregnancy, which alleviates endometrial fibrosis and enhances the regeneration competence in endometrial tissues. Abbreviation: Exo: exosome; MSC: mesenchymal stem cell; uMSC: uterus-derived SC; ADSC: adipose-derived stem cell; lncRNA: long noncoding chain RNA; VEGF: vascular endothelial growth factor; VEGFR: VEGF receptor; MMP: matrix metalloproteinase; TIMP: tissue inhibitor of metalloproteinase; α-SMA: alpha smooth muscle actin; miR: micro-RNA. ↑ represents upregulation; ↓ represents downregulation.

**Figure 7 cells-12-01018-f007:**
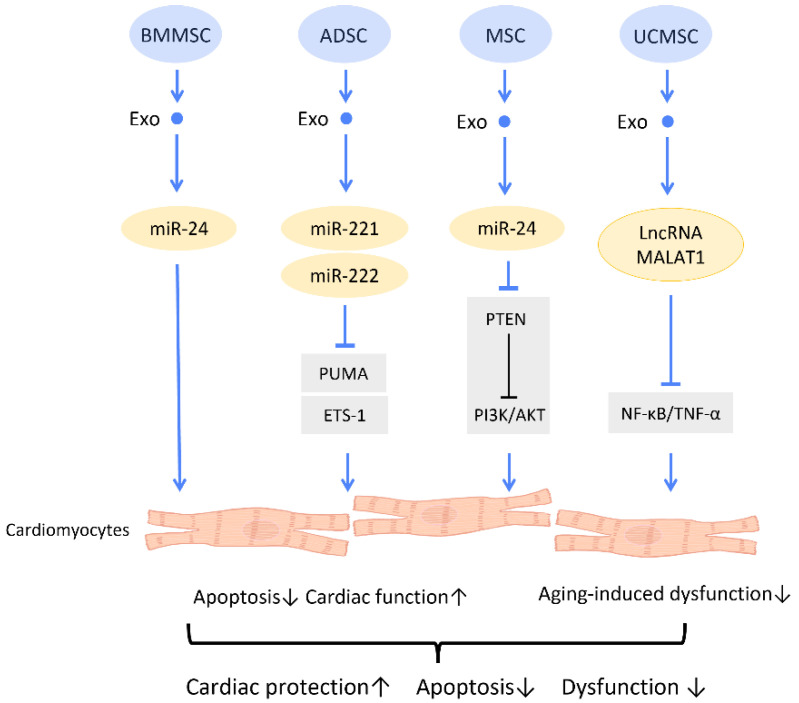
Molecular mechanisms of stem cell-derived exosomes for cardiac function recovery and protection. Stem cell-derived exosomes attenuate cardiac dysfunction and I/R injury through decreasing the oxidative damage and apoptosis of cardiac tissues. Abbreviation: Exo: exosome; SC: stem cell; MSC: mesenchymal stem cell; ADSC: adipose-derived stem cell; BMMSC: bone marrow-derived MSC; UCMSC: umbilical cord MSC; miR: micro-RNA; ROS: reactive oxygen species. ↑ represents upregulation; ↓ represents downregulation.

**Figure 8 cells-12-01018-f008:**
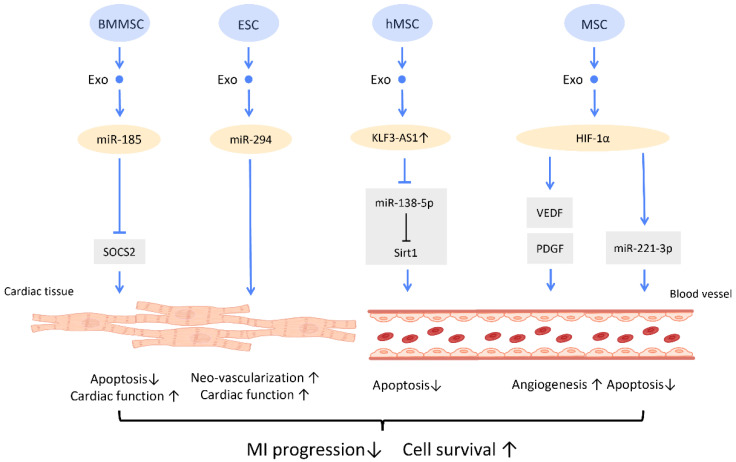
Molecular mechanisms of stem cell-derived exosomes for recovery from myocardial infarction. Molecular mechanisms of stem cell-derived exosomes for recovery from myocardial infarction. Stem-cell-derived exosomes are found to improve post-myocardial infraction recovery by inducing neo-vascularization and promoting myocyte survival. Abbreviation: Exo: exosome; SC: stem cell; MSC: mesenchymal stem cell; BMMSC: bone marrow-derived MSC; SOCS2: suppressor of cytokine signaling 2; hMSC: human MSC; miR: micro-RNA. ↑ represents upregulation; ↓ represents downregulation.

**Figure 9 cells-12-01018-f009:**
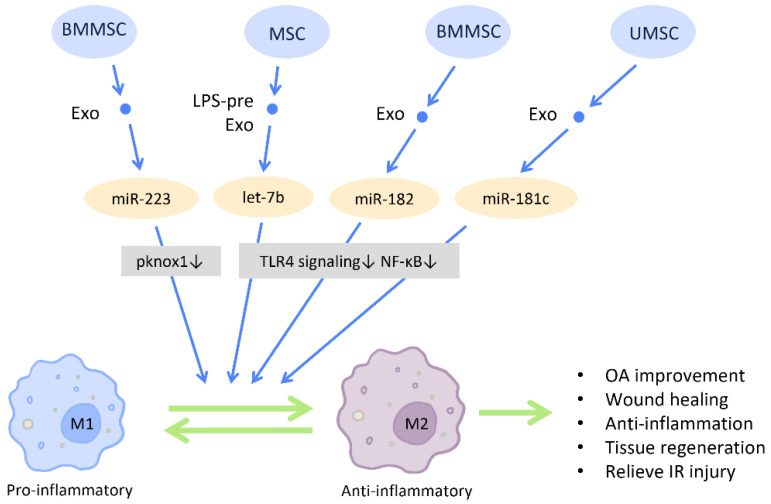
Immunomodulation by stem cell-derived exosomes in macrophage polarization. Immunomodulation by stem cell-derived exosomes in macrophage polarization. MSC-Exos have the capacity to direct macrophages from a pro-inflammatory phenotype (M1) toward an anti-inflammatory phenotype (M2), which in turn aids in the reduction of inflammation, tissue remodeling, and wound healing. Abbreviation: Exo: exosome; LPS-pre: lipopolysaccharide-preconditioned; MSC: mesenchymal stem cell; BMMSC: bone marrow MSC; UMSC: umbilical cord MSC; miR: micro-RNA; M1: M1 macrophage; M2: M2 macrophage. ↑ represents upregulation; ↓ represents downregulation.

**Figure 10 cells-12-01018-f010:**
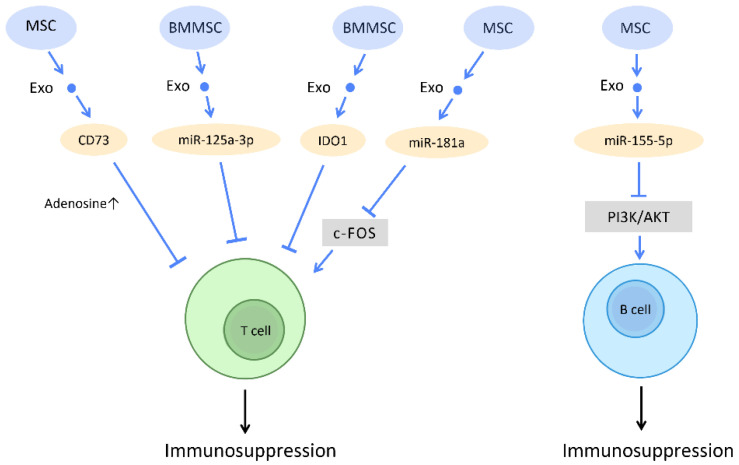
Immunosuppressive effect of stem cell-derived exosomes. Through regulating T cells and B cells function, stem cell-derived exosomes often attenuate inflammatory responses and promote a tolerogenic immune environment. Abbreviation: Exo: exosome; miR: micro-RNA; MSC: mesenchymal stem cell; BMMSC: bone marrow-derived-MSC.

**Figure 11 cells-12-01018-f011:**
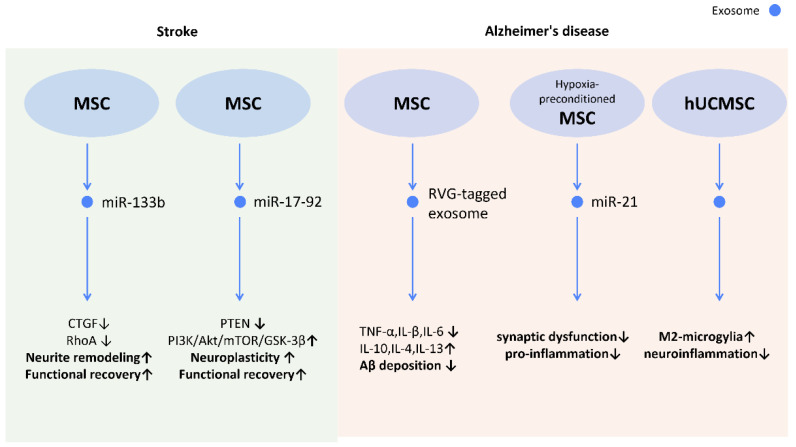
Molecular mechanisms of stem cell-derived exosomes in neurological diseases. Through shuttling molecules to neurons and glial cells, stem cell-derived exosomes possess the potential for neuronal regeneration, neuroprotection, and neuroinflammatory modulation. In stroke models, stem cell-derived exosomes promote neurite remodeling and neuroplasticity, contributing to enhanced functional recovery. In Alzheimer’s disease, stem cell-derived exosomes are shown to attenuate neuroinflammation and reduce Aβ deposition which manifested as improved memory and learning capacity. Abbreviation: CTGF: connective tissue growth factor; RhoA: ras homolog gene family member A; PTEN: phosphatase and tensin homolog; Akt: protein kinase B; mTOR; mechanistic target of rapamycin; GSK-3beta: glycogen synthase kinase 3β; hUCMSC: human umbilical cord mesenchymal stem cells. ↑ represents upregulation; ↓ represents downregulation.

## Data Availability

All data generated or analyzed during this study are included in this published article.

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
