# Peer review of "Frontier Review of the Molecular Mechanisms and Current Approaches of Stem Cell-Derived Exosomes"

_cells, 2023, doi:10.3390/cells12071018_

Round 1

Reviewer 1 Report

This review article submitted by Chen et al. described the molecular mechanisms and current approaches of stem cell-derived exosomes in a variety of physiological and pathological conditions. This topic is on the research spotlight and can be considered an exciting and new field in the biomedical research. This review is comprehensive in describing exosome biology in details. The references cited are most up to date. The illustrations are to the point and easy to understand. This review will be of great value for researchers and students to understand the state-of-the-art abut this topic. There are a few minor comments to further improve the quality of this excellent review.

1. As a review article, some of the details from prior researches can be curtained for succinct presentation. The saved space can be used to discuss the challenges in applying exsosome technology for the clinical use. What are the barriers that may prevent the wide clinical application? These discussions deserve a section prior to Conclusion.

2. The application of exosome for detection and diagnosis of human cancer can be briefly articulated as this is an emergent field.

3. In the abstract, the tone appears to be highly promising but in fact there are many road blocks in the translational research of exoxome. This needs a sentence to clarify it.

4. To improve readability, the authors should consider limiting the abbreviation used except those widely accepted terms. 

Reviewer 2 Report

This is a well-written and informative review of the use of stem cell-derived exosomes in potential clinical applications, and the underlying mechanisms by which they impart their beneficial properties.

The review is well structured and gives very detailed information to the reader. My comments are mostly that references are missing in most of the introductory paragraphs and that Figures need to be shown with better resolution and link to relevant references.

Detailed comments:

·      Abbreviations are not consistently used. For instance, MVB (line 54) is used before it is defined (line 766). In this case there is no need to define an abbreviation, since the abbreviation is used only once. Another example is the term ECM, which is defined in line 160 but later used without the abbreviation. Please check all abbreviations carefully to make sure they are defined before first used, and after that consistently use the abbreviation, rather than the full term.

·      The abbreviations of various exosomes makes sense but is not consistently used either. Please ensure that you either do not define such abbreviations (e.g. ADSC-Exos), or use them consistently throughout the text, once they are defined.

·      Many introductory paragraphs are written without referring to any literature. Please back up all such paragraphs with references to the literature, including the following

o   First two paragraphs in Section 1.2

o   Sentence covering lines 93-95.

o   Sentence ending in line 108

o   First paragraph and beginning of second paragraph Section 2.1.2

o   Sentence ending in line 193

o   Sentence ending in line 226

o   First two paragraphs in Section 2.2.4

o   First paragraph in Section 2.3

o   First two paragraphs in Section 2.3.1

o   Sentence ending in line 366

o   First paragraph in Section 2.3.2

o   First paragraph in Section 2.5

o   Sentence ending in line 500

o   Second paragraph in Section 2.5.2

o   First paragraph in Section 2.6

o   Beginning of first and second paragraphs in Section 2.6.2

o   First paragraph in Section 3.1

o   Sentence ending in line 660

o   Sentence ending in line 669

o   Sentence ending in line 711

o   Sentence ending in line 727

o   Sentence ending in line 759

·      All Figures. Please make Figures a bit larger so that the labels are legible and increase their resolution – the labels are very grainy. Also: please add relevant references to the legends, so the reader can see on which studies the information in the Figures is based.

·      Some references are not formatted correctly, e.g. Hade et al, line 199 (and references lines 452, 453, 455, 729). Please correct all such occurrences.

·      Text in line 195 refers to various growth factors as ”downstream effectors”, when they are actually usually what triggers a response (in which case they are ‘upstream’ of the signalling response). Please correct.

·      Text in line 728 “Storing unconjugated exosomes in injured tissues…” does not make sense and is not connected to the previous sentence. Please check wording.
